# Potassium-Induced Drought Tolerance of Potato by Improving Morpho-Physiological and Biochemical Attributes

Ali Asad Bahar [1], Hafiz Nazar Faried [1,*], Kashif Razzaq [1], Sami Ullah [1], Gulzar Akhtar [1], Muhammad Amin [2], Mohsin Bashir [3], Nadeem Ahmed [4], Fahad Masoud Wattoo [5], Sunny Ahmar [5], Talha Javed [6,7,*], Manzer H. Siddiqui [8], Ferdinando Branca [9] and Eldessoky S. Dessoky [10]

1. Department of Horticulture, MNS University of Agriculture, Multan 60000, Pakistan; aliasadbahar@gmail.com (A.A.B.); kashif.razzaq@mnsuam.edu.pk (K.R.); sami.ullah1@mnsuam.edu.pk (S.U.); gulzar.akhtar@mnsuam.edu.pk (G.A.)
2. Department of Horticulture Sciences, The Islamia University of Bahawalpur, Bahawalpur 63100, Pakistan; m.amin@iub.edu.pk
3. Institute of Horticultural Sciences, University of Agriculture, Faisalabad 38000, Pakistan; mohsinbashir99@gmail.com
4. Department of Plant Pathology, MNS University of Agriculture, Multan 60000, Pakistan; nadeem.ahmad@mnsuam.edu.pk
5. Department of Plant Breeding and Genetics, PMAS Arid Agriculture University, Rawalpindi 43600, Pakistan; fahad.pbg@uaar.edu.pk (F.M.W.); sunnyahmar13@gmail.com (S.A.)
6. Department of Agronomy, University of Agriculture Faisalabad, Faisalabad 38040, Pakistan
7. College of Agriculture, Fujian Agriculture and Forestry University, Fuzhou 350002, China
8. Department of Botany and Microbiology, College of Science, King Saud University, Riyadh 11451, Saudi Arabia; mhsiddiqui@ksu.edu.sa
9. Department of Agriculture, Food and Environment (Di3A), University of Catania, 95123 Catania, Italy; branca@unict.it
10. Department of Plant Genetic Transformation, Agricultural Genetic Engineering Research Institute, Agricultural Research Center, Giza 12619, Egypt; dessoky26@yahoo.com
* Correspondence: nazar.farid@mnsuam.edu.pk (H.N.F.); mtahaj@fafu.edu.cn or talhajaved54321@gmail.com (T.J.)

**Abstract:** Potato (*Solanum tuberosum*) is the third and fourth most important tuberous crop in terms of human consumption and production, respectively. However, its growth and development are affected by drought, which is an emerging threat to agriculture especially in arid and semiarid areas. Potassium (K) is a well-known macronutrient that improves the performance of crops under drought. Therefore, the present study was enacted with the aim of evaluating the impact of K fertilizer on potato crop growth, productivity, and drought tolerance under full root irrigation (FRI) and partial root irrigation (PRI) conditions. Two potato cultivars (Lady Rosetta and Hermes) were grown under normal field conditions followed by FRI and PRI applications. Potassium sulfate was applied in three doses ($T_0 = 50$ kg·ha$^{-1}$, $T_1 = 75$ kg·ha$^{-1}$, and $T_2 = 100$ kg·ha$^{-1}$). The experiment was laid out under randomized complete block design (RCBD) with split plot arrangement. The main plot was allocated to irrigation, along with a subplot to potassium and a sub-subplot to potato cultivars. The results indicated that K application significantly improved the plant growth and yield by exhibiting better performance in morpho-physiological and biochemical attributes under FRI and PRI conditions; however, a more remarkable change was noticed under PRI compared with FRI. K application alleviated drought stress regardless of cultivars. This study suggests that K application at the rate of 100 kg·ha$^{-1}$ is an effective approach for inducing drought tolerance in potato crops.

**Keywords:** *Solanum tuberosum*; potassium; drought; nutrition; antioxidants; gaseous exchange; yield

## 1. Introduction

Potato (*Solanum tuberosum* L.) is a tuberous crop and is the fourth most produced after wheat, maize, and rice. Globally, 370 million tons of potato is produced on an

area of 17.3 million ha [1]. Potato provides a sufficient amount of vitamins, proteins, carbohydrates, antioxidants, and minerals [2]. It also plays an integral role in a cropping system by producing good economic returns to the farmers and ensuring food availability to the sprawling population as a source of food, income, and employment [3]. Therefore, a substantial increase in potato production is needed [4]. Potato is an exhaustive crop and requires an excessive amount of water, i.e., 6500 $m^3$ delta of water·$ha^{-1}$ [5] and nutrients to ensure high tuber yield [6].

Drought has emerged as a serious threat to potato productivity, particularly in arid and semiarid regions [7]. The potato plants are sensitive to soil moisture stress, and an insufficiency of water supply causes a noticeable change in its growth and yield attributes [8]. The severity of water scarcity not only depends on the intensity and duration but also on the growth stages [9]. Water scarcity delays emergence, restricts root growth and nutrient uptake [10]. Additionally, a reduction in tuber number and size occurs, and resulting in yield losses [9]. Drought reduces gaseous exchange attributes [11] and biomass production in crop plants [12], while it increases the reactive oxygen species (ROS) generation, which damages DNA and causes lipid peroxidation [13]. Therefore, irrigation is always needed to produce high-yielding crops with quality attributes.

Water stress can be managed either through genetic and agronomic fortification [14] including potassium (K) application, which is a relatively inexpensive and effective strategy for inducing drought tolerance in plants [15,16]. It increases the photosynthetic rate by generating high energy status in ATP form for stimulating the plant's morpho-physiological activities, thus increasing plant growth and productivity. All plants need K, but carbohydrate-rich potato crops require it at a higher rate [17]. K supplementation increases not only the nutritive value of potato tubers [18] but also yield due to the production of a greater number of tubers [6]. Additionally, K retains a balance between endogenous antioxidant and free-radical production [18]. Furthermore, it induces drought tolerance by regulating the osmotic and turgor pressure [19]. A balanced amount of potassium keeps the plants normal, even under water-scarce conditions, and it maintains an appropriate yield and quality of produce [14].

Under different irrigationzones, all crops and genotypes respond in a diverse way to potassium application [20]. There is sufficient information available in the literature on the effect of exogenously applied potassium on growth, physiology and yield of field crops. However, little information is available regarding the interactive effect of potassium and partial root irrigation on potato cultivars in terms of their morphological, physiological, nutritive, biochemical, and yield traits under an arid environment. Keeping in view the above, two year studies were carried out on the interactive impact of potassium fertilizer and potato cultivars for inducing drought tolerance together with improving potato tuber yield and quality under full root irrigation (FRI) and moderate drought conditions developed by the partial root irrigation technique (PRI).

## 2. Materials and Methods

### 2.1. Plant Material and Experimental Site

The experiments were performed at University Research Farms, MNS-University of Agriculture, Multan, Pakistan (latitude 30°8′26.93″ N and longitude 71°26′35.43″ E) during two succeeding autumn crop seasons 2018–2019 and 2019–2020. Meteorological data of the experimental site were collected and are presented in Supplementary Figure S1. During both years, the land was prepared by constructing 30 inch ridges. Potato seeds of Lady Rosetta and Hermes (the prominent chip-making varieties in Pakistan) were planted with row–row and plant–plant spacing of 75 cm and 15 cm, respectively. Before planting, the seed potatoes were treated with a recommended dose of Emesto 24 FS (fungicide) for disinfection. Soil samples were randomly collected from the experimental field and analyzed as described by Jackson [21] (results are shown in Supplementary Table S1).

### 2.2. Potassium Fertilization and Partial Drought Application

Potassium fertilizer was applied with basal dose at the time of seedbed preparation before planting of seed potatoes. The following treatment plan was followed: $T_0$ = 50 kg/ha, $T_1$ = 75 kg/ha, and $T_2$ = 100 kg/ha. The crop was maintained under normal growing conditions until emergence. Water stress was applied by partial root irrigation technique (one side of each furrow was irrigated while the other side was kept unirrigated).

### 2.3. Biomass Attributes

Plants were harvested, for assessment of biomass attributes and later dried in an oven at 65 °C for 48 h to record dry weight.

### 2.4. Measurement of Gaseous Exchange Parameters

The top third of a fully mature, expanded, and disease-free leaf was used to measure gaseous exchange parameters (photosynthetic rate (*A*), transpiration rate (*E*), stomatal conductance (*gs*), and substomatal conductance (*Ci*)) using a portable open flow gas exchange system (CIRAS-3, Hansatech Instruments Ltd., Pentney, UK). The instrument was operated from 11:00 a.m. to 1:00 p.m. at 1200 $\mu mol \cdot m^{-2} \cdot s^{-1}$ photosynthetic photon flux concentration, 99.9 kPa atmospheric pressure, 100 $mL \cdot min^{-1}$ airflow rate, and 390 $\pm$ 5 $\mu mol \cdot mol^{-1}$ $CO_2$ concentration rate.

### 2.5. Determination of Nutrients Content

Phosphorus (P), potassium (K), and calcium (Ca) contents were determined by following earlier procedures [22]. Briefly, leaves and tubers were thoroughly washed with distilled water. After washing, samples were oven-dried at 105 °C for 2 h and later kept at 85 °C until a constant weight was achieved. About 0.5 g of each dried sample was crushed and sieved to use for the digestion process. The sample was digested by taking 5 mL of sulfuric acid in the block digester followed by the addition of distilled water to each volumetric flask up to 50 mL. Whatman No.40 filter paper was used to obtain a filtrate aliquot. K and Ca contents were analyzed using a flame photometer (BWB spectrum technologies, UK), while P contents were analyzed using a spectrophotometer (CE-7400S, Cecil Instrumentation Services Ltd., Cambridge, UK) by assisted vanadate molybdate technique.

### 2.6. Measurement of Yield Attributes

To calculate the average tuber yield (tons/ha), five randomly selected plants from each treatment block were taken and washed with tap water. The fresh weight of tubers was calculated using a digital weighing balance (OHAUS Corporation, Parsippany, NJ, USA). The average value was multiplied by the total number of plants per ha area. Average tuber length and width were measured using a digital Vernier caliper (Mitutoyo Corporation, Kanagawa, Japan).

### 2.7. Biochemical Assays

Total antioxidant activity was assessed by adopting the procedure of Mimica-Dukić et al. [23]. About 2 g of frozen tuber sample was homogenized in 5 mL of phosphate buffer solution (pH 7.0) and centrifuged at 9000 rpm for 5 min at 4 °C followed by spectrophotometric estimation at 517 nm. Superoxide dismutase (SOD) activity (U/mg of protein) was estimated by following the protocol of Štajner and Popović [24]. The enzyme extract (100 μL) was placed in test tubes, mixed well with 800 μL distilled containing having 500 μL of phosphate buffer (pH $\approx$ 5.0), 200 μL of methionine, 200 μL of Triton X, and 100 μL of NBT (Nitro blue tetrazolium), and kept in laminar airflow for 15 min under UV light. Later, riboflavin (100 μL) was supplemented, and absorbance was recorded at 560 nm. Catalase (CAT) and peroxidase (POD) activities (U/mg of protein) were analyzed using the protocol of Razzaq et al. [25]. The reaction mixture containing enzyme extract (100 μL) and $H_2O_2$ (100 μL) was used for assessing CAT, and the reading was noted at 240 nm. To measure the POD activity, 100 μL of reaction mixture containing 800 μL of phosphate buffer (pH 5.0),

100 μL of H$_2$O$_2$, and 100 μL of guaiacol was mixed with 100 μL of enzyme extract and subjected to spectrophotometric analysis at 470 nm.

### 2.8. Statistical Analysis

The data were statistically analyzed by performing Fischer's analysis of variance (ANOVA) using statistical software DASTAT (Version 1.021, Perugia, Italy). Tukey's post hoc test was used for comparing interaction means at a probability level of 0.05.

## 3. Results

### 3.1. Biomass Attributes

Supplementation of potassium fertilizer significantly ($p \leq 0.05$) influenced the growth attributes, i.e., shoot length (SL), root length (RL), shoot fresh weight (SFW), root fresh weight (RFW), shoot dry weight (SDW), and root dry weight (RDW). During both cropping seasons (2018–2019 and 2019–2020), potassium application at the rate of 100 kg·ha$^{-1}$ demonstrated the highest biomass accumulation compared to the other two treatments, regardless of irrigation treatments. Overall, Hermes performed well compared to Lady Rosetta under FRI followed by PRI (Tables 1 and 2).

**Table 1.** Morphological traits of potato cultivars supplemented with potassium fertilizer at 50, 75, and 100 kg·ha$^{-1}$ under full root irrigation (FRI) and partial root irrigation (PRI). Tukey's post hoc test was used for comparing interaction means at a probability level of 0.05.

| Irrigation Zone | Potassium Doses (kg·ha$^{-1}$) | Season 1 (2018–2019) | | | Season 2 (2019–2020) | | |
|---|---|---|---|---|---|---|---|
| | | 'Hermes' | 'Lady Rosetta' | Irrigation Means | 'Hermes' | 'Lady Rosetta' | Irrigation Means |
| | | Shoot Length (cm) | | | | | |
| FRI | 50 | 24 ± 0.37 de | 19.5 ± 0.32 f | 25.79 ± 0.33a | 20.53 ± 0.53 f | 20.93 ± 0.67 f | 25.73 ± 0.42a |
| | 75 | 27.13 ± 0.26 bc | 25.33 ± 0.24 cd | | 27.13 ± 0.26 b–d | 25.33 ± 0.24 de | |
| | 100 | 32.13 ± 0.13 a | 26.66 ± 0.66 bc | | 32.13 ± 0.13 a | 28.53 ± 0.74 bc | |
| PRI | 50 | 18 ± 0.57 f | 17.6 ± 0.33 f | 22.54 ± 0.45b | 19.26 ± 0.46 f | 16.33 ± 0.33 g | 23.85 ± 0.41b |
| | 75 | 25.33 ± 0.33 cd | 22 ± 0.57 e | | 26.66 ± 0.66 c–e | 24.4 ± 0.50 e | |
| | 100 | 28.33 ± 0.33 b | 24 ± 0.57 de | | 29.66 ± 0.33 ab | 26.8 ± 0.2 c–e | |
| | Mean (cv) | 25.89 ± 0.33 a | 22.5 ± 0.44 b | | 25.89 ± 0.39 a | 23.72 ± 0.44 b | |
| | | Root Length (cm) | | | | | |
| FRI | 50 | 15.8 ± 0.34 de | 15.13 ± 0.17 d–f | 17.25 ± 0.28a | 19 ± 0.57 d | 14.95 ± 0.52 fg | 20.46 ± 0.53a |
| | 75 | 17.86 ± 0.17 bc | 16.46 ± 0.17 b–e | | 21.86 ± 0.54 bc | 17.46 ± 0.46 de | |
| | 100 | 20.28 ± 0.35 a | 17.99 ± 0.50 bc | | 25.68 ± 0.58 a | 23.86 ± 0.56 ab | |
| PRI | 50 | 14.8 ± 0.46 ef | 13.63 ± 0.31 f | 16.43 ± 0.29b | 14.08 ± 0.59 fg | 12.86 ± 0.40 g | 17.92 ± 0.56b |
| | 75 | 17.01 ± 0.35 b–d | 16.33 ± 0.13 c–e | | 21.8 ± 0.43 d | 16.4 ± 0.80 ef | |
| | 100 | 18.6 ± 0.13 bc | 18.23 ± 0.36 b | | 22.9 ± 0.43 b | 19.53 ± 0.75 cd | |
| | Mean (cv) | 17.39 ± 0.3 a | 16.29 ± 0.27 b | | 20.8 ± 0.5 a | 17.51 ± 0.58 b | |
| | | Shoot Fresh Weight (g) | | | | | |
| FRI | 50 | 105.4 ± 1.13 d | 91.19 ± 2.34 e | 120.5 ± 1.43a | 111.6 ± 0.52 e | 97.43 ± 1.25 g | 123.7 ± 0.82a |
| | 75 | 131.4 ± 0.87 b | 111.2 ± 2.19 cd | | 129.6 ± 0.83 c | 114 ± 0.40 de | |
| | 100 | 144.8 ± 0.98 a | 139.2 ± 1.11 ab | | 148.6 ± 0.30 a | 141.2 ± 1.67 b | |
| PRI | 50 | 53.24 ± 3.60 f | 39.73 ± 0.87 g | 91.5 ± 2.11b | 102.87 ± 0.94 fg | 83.66 ± 1.57 h | 113.5 ± 1.41b |
| | 75 | 109 ± 1.73 d | 96.66 ± 4.33 e | | 130.4 ± 0.91 c | 104.13 ± 1.63 f | |
| | 100 | 133.07 ± 1.26 b | 117.5 ± 0.87 c | | 140.5 ± 1.18 d | 119.46 ± 2.24 d | |
| | Mean (cv) | 112.8 ± 1.59 a | 99.24 ± 1.95 b | | 127.2 ± 0.78 a | 109.9 ± 1.46 b | |
| | | Root Fresh Weight (g) | | | | | |
| FRI | 50 | 8 ± 0.13 d–f | 6.09 ± 0.21 g | 9.15 ± 0.19a | 14.09 ± 0.23 cd | 11.78 ± 0.26 e | 15.03 ± 0.26a |
| | 75 | 9.87 ± 0.11 bc | 8.46 ± 0.26 c–f | | 16.86 ± 0.24 b | 12.86 ± 0.23 de | |
| | 100 | 12.16 ± 0.18 a | 10.37 ± 0.31 b | | 19.33 ± 0.29 a | 15.26 ± 0.35 c | |

**Table 1.** *Cont.*

| Irrigation Zone | Potassium Doses (kg·ha⁻¹) | Season 1 (2018–2019) | | | Season 2 (2019–2020) | | |
|---|---|---|---|---|---|---|---|
| | | 'Hermes' | 'Lady Rosetta' | Irrigation Means | 'Hermes' | 'Lady Rosetta' | Irrigation Means |
| PRI | 50 | 7.67 ± 0.18 f | 3.76 ± 0.33 h | 7.96 ± 0.27b | 10.03 ± 0.27 f | 7.37 ± 0.26 g | 12.47 ± 0.29b |
| | 75 | 9.14 ± 0.24 b–d | 7.70 ± 0.31 ef | | 13.26 ± 0.29 de | 11.69 ± 0.20 e | |
| | 100 | 10.37 ± 0.29 b | 9.12 ± 0.27 b–e | | 17.53 ± 0.24 b | 14.95 ± 0.53 c | |
| | Mean (cv) | 9.53 ± 0.18 a | 7.58 ± 0.28 b | | 15.18 ± 0.26 a | 12.31 ± 0.3 b | |
| Shoot Dry Weight (g) | | | | | | | |
| FRI | 50 | 9.73 ± 0.27 c–e | 6.18 ± 0.06 gh | 10.25 ± 0.19a | 12.81 ± 0.40 f | 10.33 ± 0.36 g | 16 ± 0.40a |
| | 75 | 11.01 ± 0.24 bc | 8.02 ± 0.27 ef | | 16.71 ± 0.40 cd | 14.25 ± 0.43 ef | |
| | 100 | 13.45 ± 0.20 a | 13.13 ± 0.12 a | | 22.14 ± 0.45 a | 20.10 ± 0.37 ab | |
| PRI | 50 | 7.62 ± 0.29 fg | 5.04 ± 0.31 h | 8.27 ± 0.31b | 9.68 ± 0.40 g | 6.84 ± 0.42 h | 13.2 ± 0.36b |
| | 75 | 9 ± 0.24 d–f | 6.04 ± 0.33 gh | | 15.3 ± 0.20 de | 10.45 ± 0.29 g | |
| | 100 | 11.96 ± 0.26 ab | 10 ± 0.45 cd | | 19.1 ± 0.41 b | 18.05 ± 0.44 bc | |
| | Mean (cv) | 10.46 ± 0.25 a | 8.06 ± 0.25 b | | 15.95 ± 0.37 a | 13.33 ± 0.38 b | |
| Root Dry Weight (g) | | | | | | | |
| FRI | 50 | 1.05 ± 0.01 h | 1.37 ± 0.05 g | 2.06 ± 0.03a | 2.1 ± 0.04 d | 1.31 ± 0.02 gh | 2.12 ± 0.03a |
| | 75 | 2.02 ± 0.08 d | 1.72 ± 0.06 ef | | 2.51 ± 0.03 c | 1.65 ± 0.05 ef | |
| | 100 | 3.31 ± 0.01 a | 2.89 ± 0.00 b | | 3.25 ± 0.04 a | 1.90 ± 0.02 de | |
| PRI | 50 | 1.44 ± 0.08 fg | 0.72 ± 0.01 i | 1.59 ± 0.04b | 1.71 ± 0.03 ef | 1.04 ± 0.04 h | 2.03 ± 0.03b |
| | 75 | 1.72 ± 0.06 ef | 1.48 ± 0.01 fg | | 2.81 ± 0.04 b | 1.58 ± 0.02 fg | |
| | 100 | 2.39 ± 0.07 c | 1.79 ± 0.03 de | | 3.06 ± 0.06 ab | 2.02 ± 0.04 d | |
| | Mean (cv) | 1.98 ± 0.05 a | 1.66 ± 0.02 b | | 2.57 ± 0.04 a | 1.58 ± 0.03 b | |

To differentiate the lettering of interaction means, superscript has been used.

During the first crop season, significant ($p \leq 0.05$) increases in SL (1.33- and 1.36-fold), RL (1.28- and 1.18-fold), SFW (1.37- and 1.52-fold), RFW (1.52- and 1.70-fold), SDW (1.38- and 2.12-fold), and RDW (3.15- and 2.23-fold) were recorded in Hermes and Lady Rosetta, respectively. Moreover, during the second crop season, the same pattern was depicted in Hermes and Lady Rosetta with increments in SL (1.56- and 1.26-fold), RL (1.35- and 1.57-fold), SFW (1.33- and 1.44-fold), RFW (1.37- and 1.18-fold), SDW (1.72- and 1.82-fold), and RDW (1.54- and 1.44-fold), respectively, under FRI at 100 kg/ha $K_2O$ as compared to 50 kg/ha (Table 1).

### 3.2. Physiological Traits

Gaseous exchange attributes were significantly ($p \leq 0.05$) influenced by potassium fertilizer supplementation during both years. Exposure of potato plants to PRI caused a maximum reduction in physiological traits (Pn, E, gs, Ci) in comparison to FRI (Table 2). During the first crop season, under FRI, increases in Pn (80% and 97%), E (57% and 38%), gs (30% and 22%), and Ci (18% and 16%) were noted at 100 kg/ha $K_2O$ as compared to 50 kg/ha in both varieties (Hermes and Lady Rosetta). Similarly, during the second year, the same trend was observed in both varieties (Hermes and Lady Rosetta) with increments in Pn (54% and 74%), E (24% and 38%), gs (36% and 22%), and Ci (17% and 11%), respectively. Regardless, Hermes showed a better response than Lady Rosetta in terms of the improvement in physiological attributes under both irrigation schemes (Table 2).

**Table 2.** Physiological attributes of potato cultivars supplemented with potassium fertilizer at 50, 75, and 100 kg·ha$^{-1}$ under full root irrigation (FRI) and partial root irrigation (PRI). Tukey's post hoc test was used for comparing interaction means at a probability level of 0.05.

| Irrigation Treatment | Potassium Doses (kg·ha$^{-1}$) | Season 1 (2018–2019) | | | Season 2 (2019–2020) | | |
|---|---|---|---|---|---|---|---|
| | | 'Hermes' | 'Lady Rosetta' | Irrigation Means | 'Hermes' | 'Lady Rosetta' | Irrigation Means |
| | | Photosynthetic rate (µmol CO$_2$·m$^{-2}$·s$^{-1}$) | | | | | |
| FRI | 50 | 9.49 ± 0.47 [f] | 7.96 ± 0.45 [fg] | 12.66 ± 0.32a | 13.55 ± 0.31 [ef] | 9.16 ± 0.40 [g] | 15.32 ± 0.29a |
| | 75 | 14.01 ± 0.30 [bc] | 11.72 ± 0.30 [e] | | 18.59 ± 0.24 [bc] | 13.72 ± 0.69 [ef] | |
| | 100 | 17.10 ± 0.18 [a] | 15.7 ± 0.26 [ab] | | 20.92 ± 0.03 [a] | 16.01 ± 0.07 [d] | |
| PRI | 50 | 8.96 ± 0.37 [f] | 6.87 ± 0.03 [g] | 11.39 ± 0.18b | 9.60 ± 0.31 [g] | 7.70 ± 0.40 [g] | 13.56 ± 0.29b |
| | 75 | 12 ± 0.003 [de] | 11.47 ± 0.23 [e] | | 16.65 ± 0.26 [cd] | 12.56 ± 0.62 [f] | |
| | 100 | 15.43 ± 0.33 [ab] | 13.66 ± 0.16 [cd] | | 20 ± 0.09 [ab] | 14.86 ± 0.06 [de] | |
| | Mean (cv) | 12.83 ± 0.27 [a] | 11.23 ± 0.23 [b] | | 16.55 ± 0.20 [a] | 12.33 ± 0.37 [b] | |
| | | Transpiration rate (mmol H$_2$O·m$^{-2}$·s$^{-1}$) | | | | | |
| FRI | 50 | 2.10 ± 0.006 [f–h] | 1.97 ± 0.008 [h] | 2.49 ± 0.02a | 4.04 ± 0.03 [cd] | 3.18 ± 0.06 [e] | 4.21 ± 0.04a |
| | 75 | 2.58 ± 0.05 [cd] | 2.47 ± 0.01 [de] | | 4.18 ± 0.04 [bc] | 3.89 ± 0.06 [d] | |
| | 100 | 3.13 ± 0.06 [a] | 2.74 ± 0.01 [c] | | 5.05 ± 0.05 [a] | 4.41 ± 0.03 [b] | |
| PRI | 50 | 2.04 ± 0.02 [gh] | 2 ± 0.01 [h] | 2.26 ± 0.023b | 3.25 ± 0.02 [e] | 3.11 ± 0.001 [e] | 3.79 ± 0.03b |
| | 75 | 2.22 ± 0.008 [fg] | 2 ± 0.02 [h] | | 4.18 ± 0.02 [cd] | 3.31 ± 0.02 [e] | |
| | 100 | 3.09 ± 0.06 [b] | 2.26 ± 0.02 [ef] | | 4.86 ± 0.06 [a] | 3.98 ± 0.06 [cd] | |
| | Mean (cv) | 2.52 ± 0.03 [a] | 2.24 ± 0.01 [b] | | 4.26 ± 0.03 [a] | 3.64 ± 0.03 [b] | |
| | | Stomatal Conductance (mmol H$_2$O·m$^{-2}$·s$^{-1}$) | | | | | |
| FRI | 50 | 110.1 ± 1.12 [de] | 97.9 ± 0.16 [g] | 118.2 ± 0.85a | 141.5 ± 1.22 [d] | 113.2 ± 1.17 [f] | 149.2 ± 1.18a |
| | 75 | 125.5 ± 1.01 [c] | 112.1 ± 1.35 [d] | | 182.1 ± 1.54 [b] | 126.7 ± 1.36 [e] | |
| | 100 | 144.2 ± 1.10 [a] | 119.8 ± 0.40 [c] | | 193.6 ± 1.10 [a] | 138.6 ± 0.73 [d] | |
| PRI | 50 | 105.4 ± 1.49 [e] | 95.7 ± 0.34 [g] | 109.4 ± 0.91b | 122.8 ± 0.58 [e] | 105.9 ± 1.54 [g] | 136.2 ± 1.35b |
| | 75 | 120.1 ± 1.61 [c] | 98.7 ± 0.29 [fg] | | 160.1 ± 1.23 [c] | 123.8 ± 2.62 [e] | |
| | 100 | 132.8 ± 1.05 [b] | 104.2 ± 0.73 [ef] | | 176.7 ± 0.58 [b] | 128.2 ± 1.60 [e] | |
| | Mean (cv) | 123 ± 1.23 [a] | 105 ± 0.54 [b] | | 163 ± 1.04 [a] | 123 ± 1.5 [b] | |
| | | Sub-stomatal conductance (µmol H$_2$O·m$^{-2}$·s$^{-1}$) | | | | | |
| FRI | 50 | 152.4 ± 0.67 [d] | 131.2 ± 0.62 [g] | 154.2 ± 0.70a | 200.3 ± 1.01 [e] | 181.09 ± 0.87 [g] | 203.9 ± 0.82a |
| | 75 | 165.3 ± 0.91 [c] | 144.1 ± 0.69 [ef] | | 214.4 ± 0.55 [c] | 190.3 ± 0.68 [f] | |
| | 100 | 180.3 ± 0.72 [a] | 152 ± 0.64 [d] | | 235.9 ± 0.98 [a] | 201.7 ± 0.88 [de] | |
| PRI | 50 | 133 ± 0.57 [g] | 121.2 ± 0.62 [i] | 139.5 ± 0.79b | 186.3 ± 1.64 [fg] | 172.2 ± 0.30 [h] | 195.9 ± 1.05b |
| | 75 | 145.3 ± 0.91 [e] | 126.7 ± 0.96 [h] | | 208.2 ± 0.58 [cd] | 182.9 ± 0.54 [g] | |
| | 100 | 170.1 ± 0.62 [cd] | 141.1 ± 1.09 [f] | | 224.8 ± 0.90 [b] | 201.4 ± 2.36 [de] | |
| | Mean (cv) | 157.7 ± 0.73 [a] | 136 ± 0.77 [b] | | 211.6 ± 0.94 [a] | 188.2 ± 0.93 [b] | |

To differentiate the lettering of interaction means, superscript has been used.

### 3.3. Nutritive Attributes

In our experiment, a significant decline in nutrient (P, K, Ca) accumulation was detected due to water insufficiency. However, exogenous potassium (K) supplementation improved nutrient uptake and transport in both irrigation zones, but markedly favored FRI as compared to PRI (Figures 1 and 2). These attributes were maximum with a potassium application of 100 kg·ha$^{-1}$, except for calcium, when they were maximum at 75 kg·ha$^{-1}$ application. Moreover, Hermes accumulated more nutrients compared to Lady Rosetta.

During the first year, a noticeable increase was depicted with 100 kg·ha$^{-1}$ K$_2$O application in the contents of P in leaves (58% and 4%) and tubers (30% and 37%), K in leaves (23% and 25%) and tubers (15% and 33%), and Ca in leaves (17% and 7.3%) and tubers (48% and 21%) as compared to 50 kg·ha$^{-1}$ in both tested potato varieties (Hermes and Lady Rosetta, respectively). During the second year, the same scenario of improvement in mineral uptake was detected, i.e., increases in the contents of P in leaves (49% and 20%) and tubers (40% and 24%), K in leaves (31% and 72%) and tubers (38% and 41%), and Ca in

leaves (10% and 16%) and tubers (44% and 5.9%) in Hermes and Lady Rosetta, respectively (Figures 1 and 2).

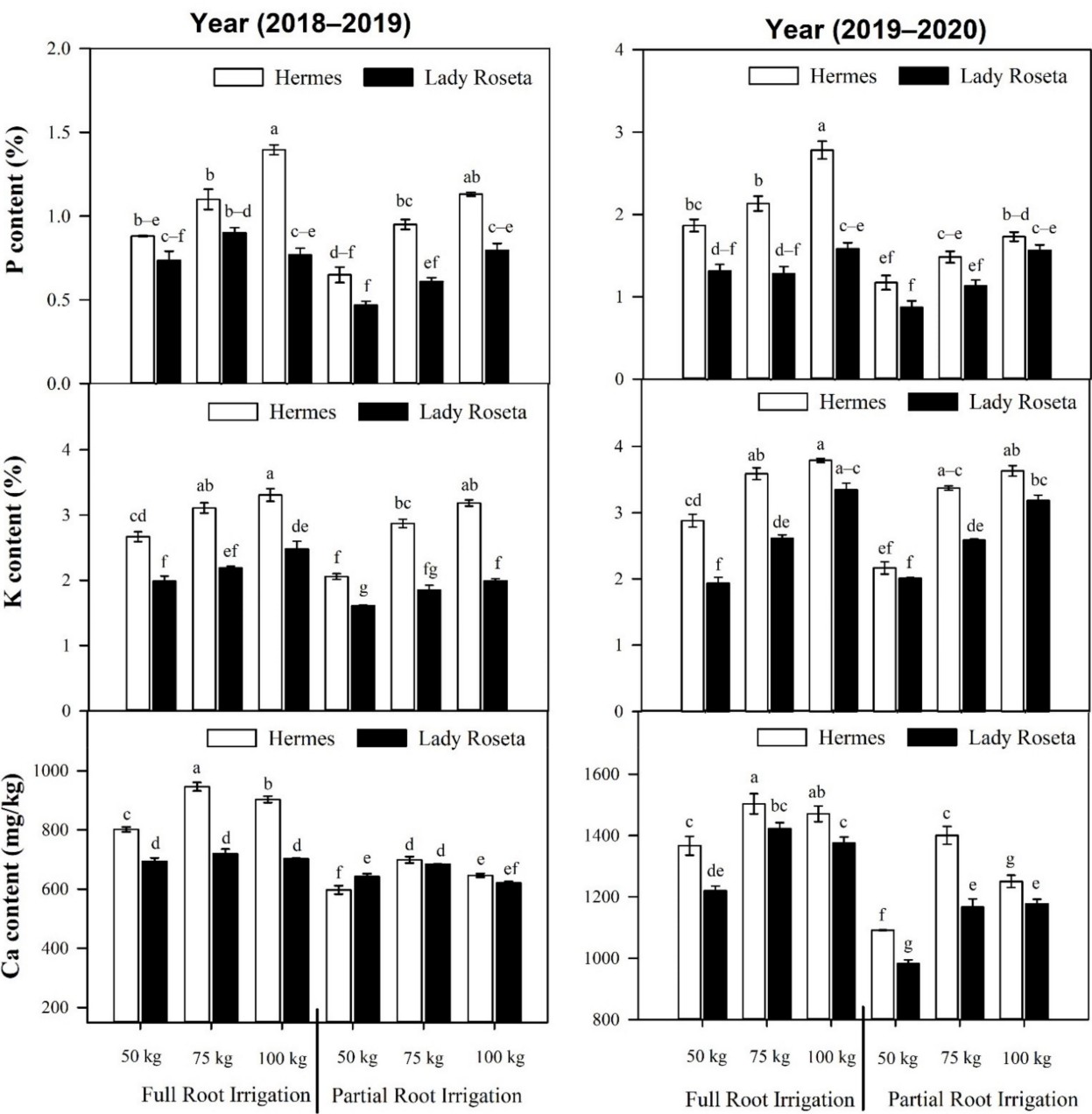

**Figure 1.** Nutritive leaf attributes of potato cultivars supplemented with potassium fertilizer at 50, 75, and 100 kg·ha$^{-1}$ under full and partial root irrigation. Graph bars delineated with different letters are significant. Vertical bars show the standard error. Tukey's post hoc test was used for comparing interaction means at a probability level of 0.05.

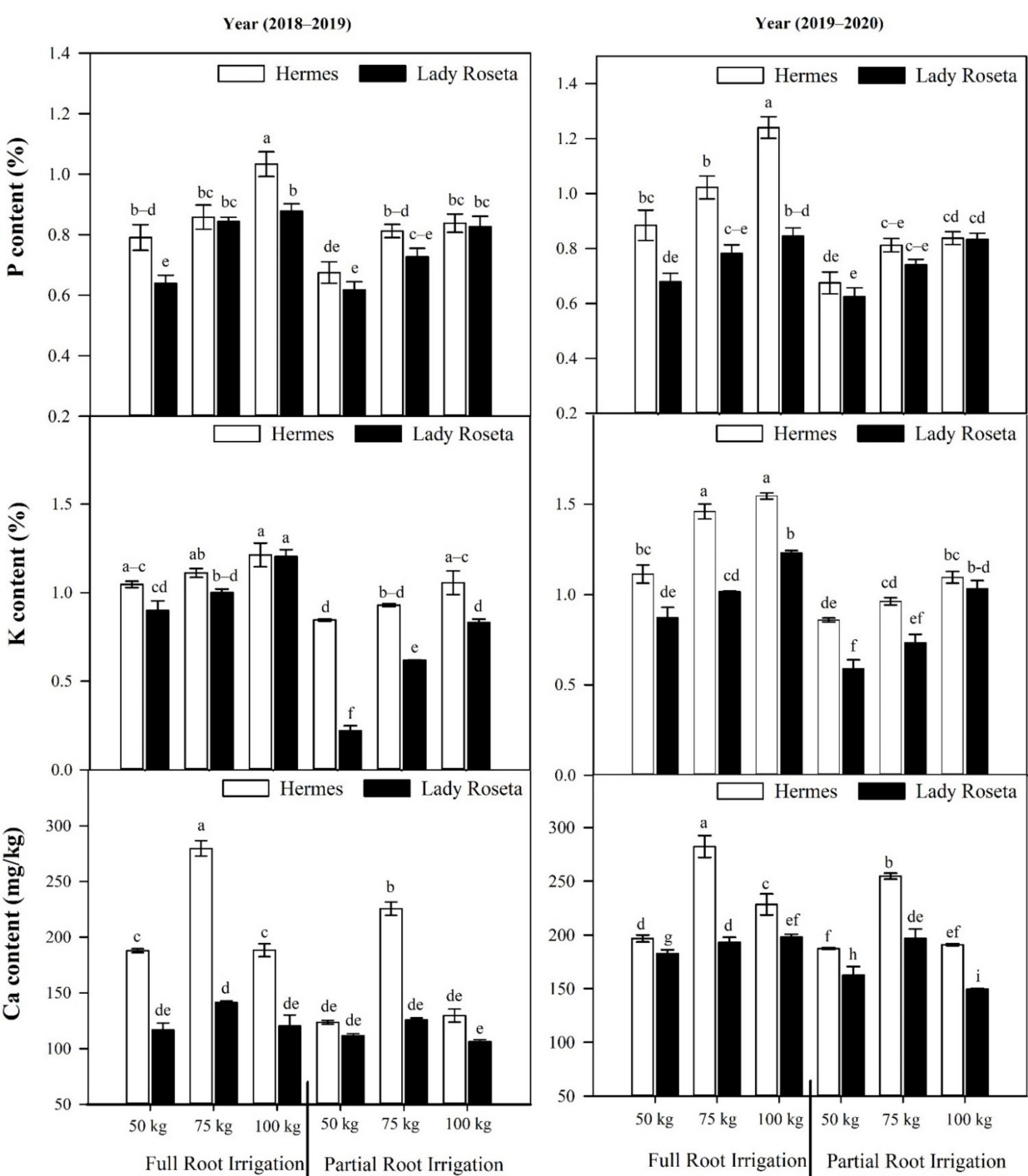

**Figure 2.** Nutritive tuber attributes of potato cultivars supplemented with potassium fertilizer at 50, 75, and 100 kg·ha$^{-1}$ under full and partial root irrigation. Graph bars delineated with different letters are significant. Vertical bars show the standard error. Tukey's post hoc test was used for comparing interaction means at a probability level of 0.05.

### 3.4. Activities/Enzymatic Antioxidants

Exogenous K application significantly ($p \leq 0.05$) improved the biochemical traits of potato cultivars. Potassium caused a significant increase in antioxidant scavenging activity (ASA) evaluated as a function of the DPPH radical-scavenging activity (Figure 3). Enzymatic activities (SOD, CAT, and POX) were also improved by potassium application (100 kg·ha$^{-1}$ compared to 50 kg·ha$^{-1}$) (Figure 4).

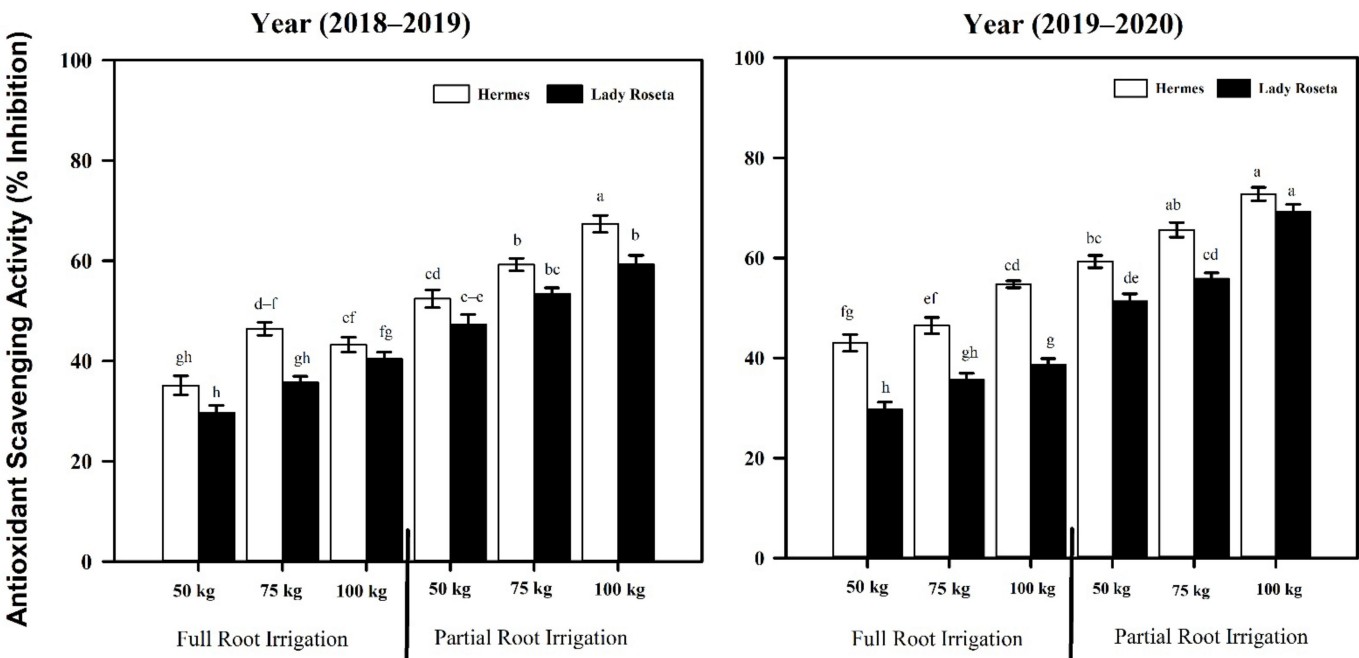

**Figure 3.** Antioxidant scavenging activity of potato cultivars supplemented with potassium fertilizer at 50, 75, and 100 kg·ha$^{-1}$ under full and partial root irrigation. Graph bars with different letters are significant. Vertical bars show the standard error. Tukey's post hoc test was used for comparing interaction means at a probability level of 0.05.

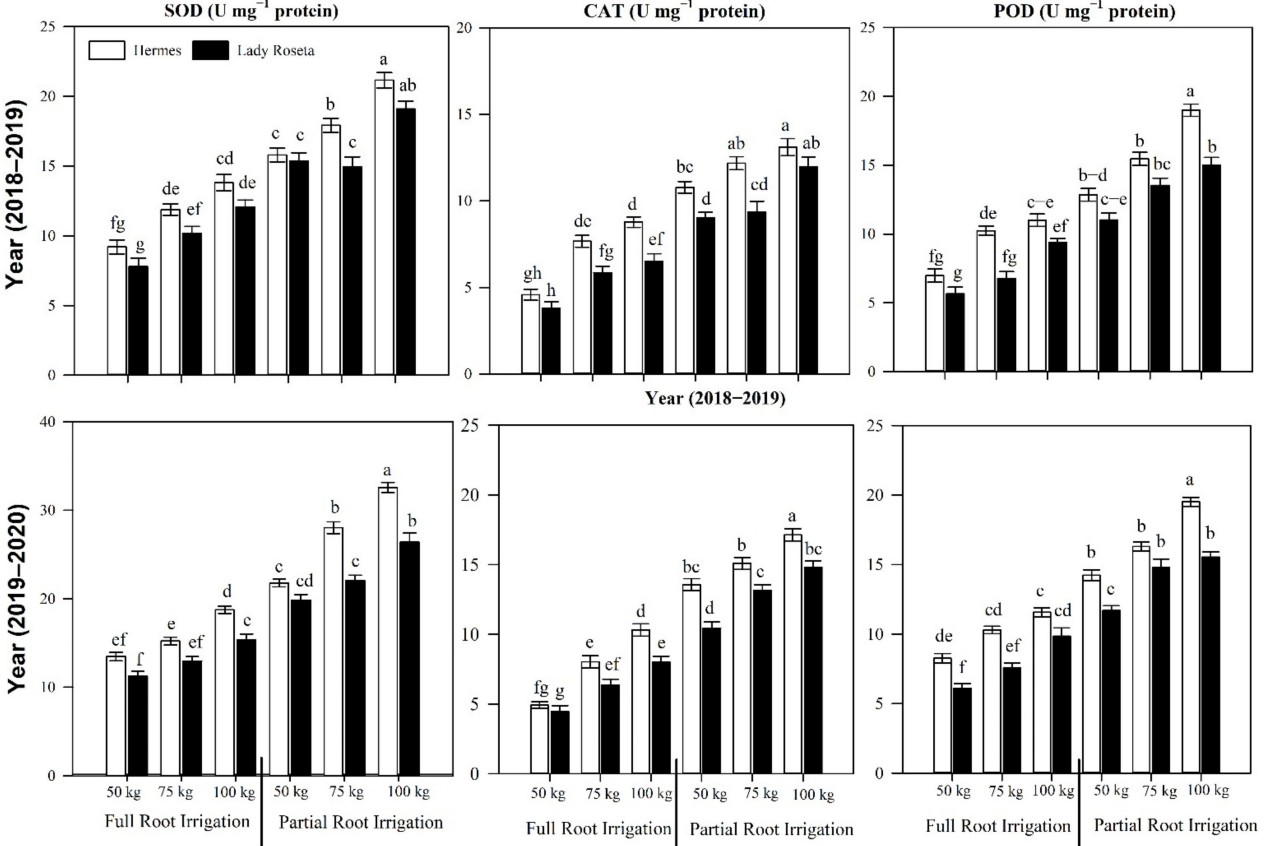

**Figure 4.** Enzymatic antioxidant activities of potato cultivars supplemented with potassium fertilizer at 50, 75, and 100 kg·ha$^{-1}$ under full and partial root irrigation. Graph bars with different letters are significant. Vertical bars show the standard error. Tukey's post hoc test was used for comparing interaction means at a probability level of 0.05.

Additionally, the plants exposed to PRI resulted in a significant increase in ASA and activities of enzymatic antioxidants with 100 kg/ha potassium application compared to FRI conditions. During the first crop season, the highest increase in ASA was noted in Hermes (1.28-fold) followed by Lady Rosetta (1.25-fold) with 100 kg/ha potassium application under PRI in comparison to 50 kg/ha. During the second year, under PRI conditions, the highest increase in ASA was exhibited by Lady Rosetta (1.34-fold) followed by Hermes (1.22) with 100 kg/ha potassium application relative to 50 kg/ha. An increase in enzymatic antioxidants (SOD, CAT, POX) was also recorded during both cropping years. During the first year, increases in SOD (1.33- and 1.24-fold), CAT (1.21- and 1.32-fold), and POX (1.47- and 1.36-fold) activities were noted in Hermes and Lady Rosetta, grown with 100 $kg \cdot ha^{-1}$ potassium under PRI as compared to 50 $kg \cdot ha^{-1}$. Similarly, during the second year, maximum SOD (1.49- and 1.33-fold), CAT (1.26- and 1.41-fold), and POX (1.37- and 1.32-fold) activities were observed in Hermes and Lady Rosetta under PRI conditions with 100 $kg \cdot ha^{-1}$ relative to a minimum application (50 $kg \cdot ha^{-1}$). Overall, the ASA and enzymatic activities helped prevent the plants from oxidative damage, and this effect was higher in Hermes than in Lady Rosetta (Figures 3 and 4).

### 3.5. Yield and Yield Elements

Potassium application significantly ($p \leq 0.05$) influenced yield and its attributes, i.e., tuber fresh weight (TFW), tuber length (TL), tuber width (TW), and tuber yield (TY), in tested potato cultivars (Table 3). The potato plants grown with 100 $kg \cdot ha^{-1}$ $K_2O$ under both irrigation schemes delineated the highest increase in yield and its components followed by 75 $kg \cdot ha^{-1}$ and 50 $kg \cdot ha^{-1}$. Amazingly, yield was higher under PRI as compared to FRI. During the first year, under PRI, the higher yield was noted in Hermes (2.24-fold) than Lady Rosetta (2.09-fold) with 100 $kg-ha^{-1}$ $K_2O$ compared with a minimum application (50 $kg \cdot ha^{-1}$ $K_2O$). Similarly, during the second year, higher yield was observed in Hermes (2.18-fold) than Lady Rosetta (1.87-fold) with 100 $kg \cdot ha^{-1}$ $K_2O$ compared to 50 $kg \cdot ha^{-1}$ $K_2O$. Conclusively, Hermes showed higher performance in attaining yield and its components over Lady Rosetta (Table 3).

**Table 3.** Yield attributes of potato cultivars supplemented with potassium fertilizer at 50, 75, and 100 $kg \cdot ha^{-1}$ under full root irrigation (FRI) and partial root irrigation (PRI). Tukey's post hoc test was used for comparing interaction means at a probability level of 0.05.

| Irrigation Treatments | Potassium Doses ($kg \cdot ha^{-1}$) | Season 1 (2018–2019) | | | Season 2 (2019–2020) | | |
|---|---|---|---|---|---|---|---|
| | | 'Hermes' | 'Lady Rosetta' | Irrigation Means | 'Hermes' | 'Lady Rosetta' | Irrigation Means |
| | | Tuber Fresh Weight (g) | | | | | |
| FRI | 50 | 76.65 ± 1.70 [ef] | 52.85 ± 1.44 [h] | | 82.15 ± 1.12 [f] | 70.22 ± 1.31 [g] | |
| | 75 | 93.24 ± 1.34 [c] | 63.54 ± 1.07 [g] | 78.82 ± 1.35b | 97.57 ± 0.82 [de] | 90.42 ± 1.63 [e] | 97.50 ± 1.3b |
| | 100 | 102.07 ± 1.13 [ab] | 84.57 ± 1.47 [d] | | 132.06 ± 1.24 [ab] | 115.62 ± 1.68 [c] | |
| PRI | 50 | 72.64 ± 1.81 [f] | 52.44 ± 1.22 [h] | | 75.74 ± 1.23 [fg] | 54.01 ± 1.24 [h] | |
| | 75 | 96.57 ± 1.51 [bc] | 80.05 ± 1.71 [de] | 83.50 ± 1.42a | 101.31 ± 2.13 [d] | 98.09 ± 0.84 [d] | 99.26 ± 1.31a |
| | 100 | 105.23 ± 1.27 [a] | 94.11 ± 1.02 [c] | | 137.06 ± 1.24 [a] | 129.39 ± 1.18 [b] | |
| | Mean (cv) | 91.06 ± 1.46 [a] | 71.24 ± 1.32 [b] | | 104.31 ± 1.29 [a] | 92.95 ± 1.31 [b] | |
| | | Tuber Length (cm) | | | | | |
| FRI | 50 | 5.73 ± 0.21 [a–d] | 4.14 ± 0.01 [f] | | 5.42 ± 0.24 [de] | 5.15 ± 0.02 [ef] | |
| | 75 | 6.06 ± 0.09 [a–c] | 5.42 ± 0.13 [b–d] | 5.55 ± 0.10a | 6.22 ± 0.26 [b–d] | 6 ± 0.06 [cd] | 6 ± 0.13a |
| | 100 | 6.16 ± 0.07 [ab] | 5.83 ± 0.13 [a–c] | | 6.90 ± 0.13 [ab] | 6.32 ± 0.10 [a–c] | |
| PRI | 50 | 5 ± 0.14 [de] | 4.49 ± 0.03 [ef] | | 4.89 ± 0.16 [ef] | 4.37 ± 0.22 [f] | |
| | 75 | 5.57 ± 0.25 [a–d] | 5.24 ± 0.08 [c–e] | 5.43 ± 0.12b | 6.27 ± 0.17 [a–c] | 6.18 ± 0.20 [b–d] | 5.80 ± 0.14b |
| | 100 | 6.39 ± 0.12 [a] | 5.91 ± 0.15 [a–c] | | 7.09 ± 0.10 [a] | 6.06 ± 0.01 [cd] | |
| | Mean (cv) | 5.81 ± 0.14 [a] | 5.17 ± 0.08 [b] | | 6.13 ± 0.18 [a] | 5.68 ± 0.10 [b] | |

**Table 3.** *Cont.*

| Irrigation Treatments | Potassium Doses (kg·ha⁻¹) | Season 1 (2018–2019) | | | Season 2 (2019–2020) | | |
|---|---|---|---|---|---|---|---|
| | | 'Hermes' | 'Lady Rosetta' | Irrigation Means | 'Hermes' | 'Lady Rosetta' | Irrigation Means |
| | | Tuber Width (cm) | | | | | |
| FRI | 50 | 3.73 ± 0.06 [e] | 4.14 ± 0.01 [b–d] | 4.12 ± 0.04a | 4.01 ± 0.14 [d] | 4.41 ± 0.07 [b–d] | 4.38 ± 0.09a |
| | 75 | 4.02 ± 0.03 [c–e] | 4.27 ± 0.04 [a–c] | | 4.33 ± 0.14 [b–d] | 4.57 ± 0.01 [a–c] | |
| | 100 | 4.23 ± 0.08 [a–c] | 4.35 ± 0.06 [ab] | | 4.29 ± 0.07 [b–d] | 4.72 ± 0.15 [ab] | |
| PRI | 50 | 3.13 ± 0.02 [f] | 3.83 ± 0.03 [de] | 4.03 ± 0.02b | 3.16 ± 0.02 [e] | 4.53 ± 0.04 [a–c] | 4.22 ± 0.04b |
| | 75 | 4.20 ± 0.01 [bc] | 4.13 ± 0.02 [b–d] | | 4.03 ± 0.11 [d] | 4.13 ± 0.02 [cd] | |
| | 100 | 4.36 ± 0.03 [ab] | 4.53 ± 0.04 [a] | | 4.60 ± 0.04 [a–c] | 4.89 ± 0.04 [a] | |
| | Mean (cv) | 3.94 ± 0.03 [b] | 4.20 ± 0.03 [a] | | 4.07 ± 0.09 [b] | 4.54 ± 0.06 [a] | |
| | | Tuber Yield (tons·ha⁻¹) | | | | | |
| FRI | 50 | 15.56 ± 0.51 [g] | 12.81 ± 0.03 [hi] | 19.81 ± 0.35b | 19.56 ± 0.91 [fg] | 16.29 ± 1.00 [gh] | 24.25 ± 0.65a |
| | 75 | 22.40 ± 0.61 [d] | 19.24 ± 0.38 [f] | | 28.09 ± 0.31 [c] | 20.79 ± 0.68 [ef] | |
| | 100 | 27.49 ± 0.37 [b] | 21.39 ± 0.21 [de] | | 36.77 ± 0.53 [a] | 24.03 ± 0.48 [de] | |
| PRI | 50 | 13.61 ± 0.63 [h] | 11.79 ± 0.32 [i] | 21.12 ± 0.4a | 16.77 ± 0.81 [gh] | 13.59 ± 0.68 [h] | 25.19 ± 0.61a |
| | 75 | 25.23 ± 0.26 [c] | 20.74 ± 0.06 [ef] | | 32.73 ± 0.11 [b] | 24.24 ± 0.91 [d] | |
| | 100 | 30.63 ± 0.67 [a] | 24.76 ± 0.46 [c] | | 36.56 ± 0.60 [a] | 27.29 ± 0.60 [cd] | |
| | Mean (cv) | 22.48 ± 0.51 [a] | 18.45 ± 0.23 [b] | | 28.41 ± 0.54 [a] | 21.03 ± 0.72 [b] | |

To differentiate the lettering of interaction means, superscript has been used.

## 4. Discussion

Water scarcity is a limiting factor for crop productivity [26]. In fact, drought has emerged as a major constraint in potato production, particularly in arid and semiarid regions, affecting the crop at establishment, stolon and tuber initiation, bulking, and maturation stages [7]. During the present study, drought imposed negative impacts on the performance of potato crop by reducing the growth (SL, RL, SFW, RFW, SDW, and RDW), gaseous exchange (*Pn*, *E*, gs, and *Ci*), (Tables 1 and 2), and nutritive attributes (P, K, and Ca) (Figures 1 and 2). However, maximum potassium supplementation (100 kg·ha⁻¹) induced drought tolerance by improving growth, gaseous exchange, nutritive, antioxidant, and yield attributes. As a stress reliever, potassium mitigated drought's negative impacts by regulating or improving stomatal conductance, photosynthetic rate, and therefore potato growth by strengthening the source-to-sink relationship [27]. Potassium application improved the gaseous exchange attributes probably by regulating $CO_2$ intake and ATP synthesis [14]. This protective influence might also be due to the mitigation of the adverse impact of osmotic stress via improving chlorophyll concentration [28], protein abundance, and Rubisco activity that acted as a carboxylase, thereby reducing the ROS generation by regulating the plant's antioxidative activities [29]. The findings of this study are consistent with the results reported by [30,31]. Nutritive attributes (P, K, and Ca) were improved by exogenous potassium supplementation (100 kg·ha⁻¹) in both leaves and tubers under both irrigation zones. However, the above mineral contents were higher under FRI as compared to PRI (Figures 1 and 2). The restrictions in nutrient accumulation may be linked with reduced transpiration rate, resulting in a reduction in root absorbing power to take up nutrients [32,33]. Moreover, a positive relationship exists between root length and nutrient uptake. Plants increase their root length and surface area to acquire nutrients [34]. However, drought restricted the plant root growth, and the same reducing trend of root length was observed during the present study (Table 1). Both yield and its related traits such as TFW, TL, and TW were significantly improved by potassium doses under FRI and PRI. Surprisingly, these attributes were maximum under PRI as compared to FRI (Table 3). This improvement may be due to osmolyte accumulation, stomatal functioning, and distribution and activation of antioxidants, which improved tuber yield and quality. Another possible reason for this explanation is that increasing irrigation might result in stomatal closure that ultimately exerts negative impacts on yield and yield-related aspects in line with the find-

ings of previous reports [19], where an increased amount of water had a negative impact on yield and yield-related attributes. Additionally, the improved role of PRI in terms of yield characteristics has been reported in potato [35], tomato [36], pepper [37], eggplant [38], and sugar beet [39]. Moreover, DPPH and antioxidant enzyme activities were improved by potassium application in line with [40]. A recent study [41] confirmed that partial root irrigation enhances antioxidant activities in potato tubers. Moreover, in these studies, activities of antioxidant enzymes such as SOD, CAT, and POD were increased significantly ($p \leq 0.05$) under PRI at maximum potassium application of $100 \, \mathrm{kg \cdot ha^{-1}}$ (Figures 3 and 4). Enhanced antioxidant activities by potassium supplementation under water scarcity have been reported in potato, tomato, and eggplant [42–44]. Generally, plants use complex antioxidant defense systems to mitigate the generation of ROS and defend the plants from oxidative damage [45]. Both water scarcity and potassium deprivation result in the accumulation of ROS [14]. However, sufficient potassium application enhances antioxidative strength and allows scavenging more ROS in plants [46]. A possible explanation for the increase in antioxidant activities (SOD, CAT, and POD) might be the positive effect of K in ameliorating drought via increasing relative water content and leaf turgor [47].

## 5. Conclusions

Potassium application improved the potato plant growth and productivity under both full root irrigation (FRI) and partial root irrigation (PRI) conditions, but effects were more prominent under PRI. Conclusively, potassium application at $100 \, \mathrm{kg \cdot ha^{-1}}$ induced drought tolerance and improved potato growth and productivity, regardless of variety and irrigation treatment. Overall, Hermes performed well in all aspects as compared to Lady Rosetta. However, a higher level of K should be considered in future studies.

**Supplementary Materials:** The following are available online at https://www.mdpi.com/article/10.3390/agronomy11122573/s1: Figure S1. Meteorological data of mean temperature and mean relative humidity during 2019 and 2020 experiment periods; Table S1. Physicochemical analysis of soil.

**Author Contributions:** Conceptualization, H.N.F., T.J. and A.A.B.; methodology, H.N.F. and A.A.B.; software, S.U.; validation, H.N.F., M.A. and G.A.; formal analysis, A.A.B.; investigation, A.A.B.; resources, H.N.F. and K.R.; data curation, H.N.F. and A.A.B.; writing—original draft preparation, H.N.F. and A.A.B.; writing—review and editing, T.J., S.A., F.B., E.S.D., M.H.S., H.N.F., M.B., F.M.W. and N.A.; visualization, H.N.F. and A.A.B.; supervision, H.N.F.; project administration, H.N.F.; funding acquisition, H.N.F., M.A. and K.R. All authors have read and agreed to the published version of the manuscript.

**Funding:** Researchers Supporting Project number (RSP-2021/347), King Saud University, Riyadh, Saudi Arabia.

**Institutional Review Board Statement:** Not applicable.

**Informed Consent Statement:** Not applicable.

**Data Availability Statement:** Not applicable.

**Acknowledgments:** We gratefully acknowledge the university research farms, central lab system, and plant propagation and physiology lab for the successful execution of this research. Authors would like to extend their sincere appreciation to the Researchers Supporting Project number (RSP-2021/347), King Saud University, Riyadh, Saudi Arabia.

**Conflicts of Interest:** The authors declare no conflict of interest.

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
