# Peer review of "Potassium-Induced Drought Tolerance of Potato by Improving Morpho-Physiological and Biochemical Attributes"

_agronomy, doi:10.3390/agronomy11122573_

Round 1

Reviewer 1 Report

The manuscript presents original results of the effect of potassium fertilizer on potatoes belonging to two cultivars subjected to full root irrigation (FRI) and partial root irrigation (PRI). The results have been presented and discussed in detail and can be useful in practice. However, some detailed comments need to be considered. 

line 35: Detailed data on current potato production could be useful. The latest data should be cited. 

lines 66-67: It has been written: "Under different agro-ecological zones, all crops and genotypes response in a diverse way to potassium application [19]." A "diverse way" should be explained in detail. Please also add more references including the newest ones.

lines 67-69: It has been written: "there is a limited information on the interactive effect of potassium and potato cultivars to water scarcity for morpho, physio, nutritive, biochemical and yield traits under arid environment." What exactly does "limited information" mean? What is the greatest novelty of present research on the background of the available literature data? 

line 81: Please add the justification for the selection of Lady Roseta and Hermes varieties.

The description of subsection 2.8. Statistical analysis should be more detailed. Additionally, footnotes regarding statistical results should be provided for tables. What exactly was compared? All data for FRI and PRI?

line 150: How were folds calculated?

Figure 1 and Figure 2: Why is the Ca content in mg/kg, and K and P contents in %?

Section 5. Conclusions should be supplemented. Information is too general.

The conclusions should be more supported by the results.

Author Response

The manuscript presents original results of the effect of potassium fertilizer on potatoes belonging to two cultivars subjected to full root irrigation (FRI) and partial root irrigation (PRI). The results have been presented and discussed in detail and can be useful in practice. However, some detailed comments need to be considered. 

line 35: Detailed data on current potato production could be useful. The latest data should be cited. 

Answer to Reviewer 1: Incorporated and highlighted with red color

lines 66-67: It has been written: "Under different agro-ecological zones, all crops and genotypes response in a diverse way to potassium application [19]." A "diverse way" should be explained in detail. Please also add more references including the newest ones.

Answer to Reviewer 1: Under different agro-ecological zones, crops response vary to various potassium application. For instances, the study conducted in Saudi Arabia demonstrated increase in yield of about 78% with potassium application of 225kg/ha while 22% increase in yield observed with 150kg/ha potassium application in Pakistan.

lines 67-69: It has been written: "there is a limited information on the interactive effect of potassium and potato cultivars to water scarcity for morpho, physio, nutritive, biochemical and yield traits under arid environment." What exactly does "limited information" mean? What is the greatest novelty of present research on the background of the available literature data? 

Answer to Reviewer 1: Sufficient information available in literature on the effect of exogenously applied potassium on growth, physiology, and yield of field crops like Maiz. However, little information available regarding the interactive effect of potassium and partial root irrigation on potato cultivars for morpho, physio, nutritive, biochemical and yield traits under arid environment

line 81: Please add the justification for the selection of Lady Roseta and Hermes varieties.

Answer to Reviewer 1: Lady Roseta and Hermes are the top chip making processing varieties grown in Pakistan. Their cultivation has continuously been increasing in Punjab Province due the establishment of the Asia’s largest chip industry in Punjab by PEPSICO. Similarly, Pakistan is the 5th most affected country by climate change and declares as the water scare country. In this scenario, these varieties were selected for mentioned studies for drought tolerance induction through potassium.

The description of subsection 2.8. Statistical analysis should be more detailed. Additionally, footnotes regarding statistical results should be provided for tables. What exactly was compared? All data for FRI and PRI?

Answer to Reviewer 1: Suggested changes incorporated in the respective section. Tuckey’s post hoc test was used for comparing interaction means at a probability level of 0.05. Irrigation× potassium× varieties comparison was done both at full and partial root irrigation.

line 150: How were folds calculated?

Answer to Reviewer 1: Folds were calculated by dividing the new amount by original value (Control) i.e., if control treatment value is 2 and non-control treatment value is 8, then 8/2=4 folds.

Figure 1 and Figure 2: Why is the Ca content in mg/kg, and K and P contents in %?

Answer to Reviewer 1: Calcium contents were determined in mg/kg as the calcium contents were present in trace amount in potatoes. That’s why, it has been estimated in mg/kg unit.

Section 5. Conclusions should be supplemented. Information is too general. The conclusions should be more supported by the results.

Answer to Reviewer 1: Incorporated and highlighted with red color in the conclusion section.

Reviewer 2 Report

Agronomy Manuscript

General comments:

-In general, this manuscript has a valuable topic. The topic is scientifically sound.

-The writing style and English language is fine.

experimental design is adequate. My main concerns were the introduction, the Data presentation, and the Discussion section.                                                                                    

-There are some MAJOR comments.

Detailed comments:

-In general, please avoid using personal pronouns such as we, our results, our work and apply this rule throughout the manuscript (for example -Line# 175: In our experiment , Line # 264 our findings and more).

Abstract:

-This section is missing the direct aim of the study, please write the aim as following; The aim of this study was….

-Key words; Please add the word potassium and Drought tolerance to the keywords list.

Introduction:

The topic is very important and has a great value. I see that the introduction didn’t provide enough background about the topic and needs to be enriched,

Materials and Methods:

The experimental design was suitable and adequate to the current study.

Results:

-In general, I found that the data presentation in figures 1-3 is poorly presented and very confusing. Please present data in figures in a way that the reader can easily understand them. Also add detailed legend and specific panels for each parameter in the figures.

Discussion

This section is poorly written. I had a hard time to relate the discussion section with the corresponding data in the tables and figures, please rewrite this section and provide the appropriate citations in argument, and valuable discussion to the current results.

Conclusion:

This section is well written. It is Short but. provides a good conclusion for the study and include the significant findings. This section is supported by the results and include the author suggestion for the future work.

References:

The authors provided enough citations, But NOT up to Date.

-Please provide enough recent citations from the last 5 years research

Author Response

-In general, please avoid using personal pronouns such as we, our results, our work and apply this rule throughout the manuscript (for example -Line# 175: In our experiment, Line # 264 our findings and more).

Answer to Reviewer 2: Implemented

Abstract:

-This section is missing the direct aim of the study, please write the aim as following; The aim of this study was….

Answer to Reviewer 2: Incorporated as suggested and highlighted with red color

-Key words; Please add the word potassium and Drought tolerance to the keywords list

Answer to Reviewer 2: Incorporated as suggested and highlighted with red color

Introduction:

The topic is very important and has a great value. I see that the introduction didn’t provide enough background about the topic and needs to be enriched.

Answer to Reviewer 2: Incorporated as suggested and highlighted with red color

Materials and Methods:

The experimental design was suitable and adequate to the current study.

Results:

-In general, I found that the data presentation in figures 1-3 is poorly presented and very confusing. Please present data in figures in a way that the reader can easily understand them. Also add detailed legend and specific panels for each parameter in the figures.

Answer to Reviewer 2: Revised and updated the figures as suggested

Discussion

This section is poorly written. I had a hard time to relate the discussion section with the corresponding data in the tables and figures, please rewrite this section and provide the appropriate citations in argument, and valuable discussion to the current results.

Answer to Reviewer 2: Incorporated the updated references, improved as suggested and highlighted with red color

Conclusion:

This section is well written. It is Short but. provides a good conclusion for the study and include the significant findings. This section is supported by the results and include the author suggestion for the future work.

Answer to Reviewer 2: Improved as suggested and highlighted with red color.

References:

The authors provided enough citations, But NOT up to Date.

-Please provide enough recent citations from the last 5 years research

Answer to Reviewer 2: Incorporated as suggested and highlighted with red color

Round 2

Reviewer 1 Report

The manuscript has been improved. However, I have one additional comment. In the Conclusions, the abbreviations 'FRI' and 'PRI' should be explained.

Author Response

Comments and Suggestions for Authors

The manuscript has been improved. However, I have one additional comment. In the Conclusions, the abbreviations 'FRI' and 'PRI' should be explained.

Response: Esteemed reviewer, thank you for your appreciated comments/suggestions. We have explained the abbreviations.

Reviewer 2 Report

I reviewed the revised version. The authors dis a great job to address my comments. This manuscript is suffiently improved and now it is suitable for publication.

Author Response

Comments and Suggestions for Authors

I reviewed the revised version. The authors dis a great job to address my comments. This manuscript is suffiently improved and now it is suitable for publication.

Response: Esteemed reviewer, thank you for your appreciated comments.

This manuscript is a resubmission of an earlier submission. The following is a list of the peer review reports and author responses from that submission.